# Widow spiders alter web architecture and attractiveness in response to same-sex competition for prey and mates, and predation risk

Andreas Fischer [1✉], Yasasi Fernando [1], April Preston[1], Sarah Moniz-de-Sa[1] & Gerhard Gries[1]

Female-female competition in animals has rarely been studied. Responses of females that compete context-dependently for mates and prey, and seek safety from predators, are ideally studied with web-building spiders. Cobwebs possess unique sections for prey capture and safety, which can be quantified. We worked with *Steaoda grossa* females because their pheromone is known, and adjustments in response to mate competition could be measured. Females exposed to synthetic sex pheromone adjusted their webs, indicating a perception of intra-sexual competition via their sex pheromone. When females sequentially built their webs in settings of low and high intra-sexual competition, they adjusted their webs to increase prey capture and lower predation risk. In settings with strong mate competition, females deposited more contact pheromone components on their webs and accelerated their breakdown to mate-attractant pheromone components, essentially increasing their webs' attractiveness. We show that females respond to sexual, social and natural selection pressures originating from intra-sexual competition.

[1] Department of Biological Sciences, Simon Fraser University, Burnaby, BC, Canada. ✉email: andifischer10@outlook.com

Competition is a universal phenomenon that arises from the scarceness of resources and mates[1]. Sexual conflict originates when males and females have conflicting optimal reproductive fitness strategies, possibly leading to an evolutionary 'arms race' between males and females[2–4]. Moreover, there is a struggle not only between conspecific males and females but also among males and among females[5–7]. In this same-sex competition, males—and rarely females—compete for mates (sexual selection[8]) or for essential resources (social selection[9]). Same-sex competition is an evolutionary force shaping traits that increase reproductive fitness[8]. Surprisingly, female-female competition has received little attention[10]. Studies of evolutionary selection for secondary sexual traits, such as marked coloration, large size, or striking adornments, have prioritized males over females, possibly because these traits are generally more apparent in males than in females[10]. That competition among females can be an evolutionary force has only recently been acknowledged[11].

While females rarely directly compete for mates, "maternally-biased reproductive investment renders females more likely than males to experience intense competition for resources important for reproduction"[5,6]. Females competing for reproductive resources not directly associated with mates (social selection; contest competition[9,12]) may be subject to evolutionary selection for traits, such as larger 'weaponry', and thus improved reproductive fitness[6]. For example, female dung beetles, *Onthophagus sagittarius*, competing for animal feces as a reproductive resource, are subject to selection for larger weaponry and thus improved reproductive fitness[6]. Conspecific females that are in competition with each other over reproductive resources such as nutrients and offspring development sites must be able to sense and process information about their competitors to remain engaged in the co-evolutionary 'arms race'[12]. Communication modalities conveying such information may be visual, vibratory (acoustic or substrate-vibration), or chemical (smell or taste) in nature[13].

Chemical communication is thought to be the oldest mode of information transmission[14], and sex pheromones, serving as intraspecific sexual communication signals that facilitate mate attraction, recognition, and acceptance, greatly contribute to reproductive success and survival[15]. Sex pheromones may elicit behavioral responses or cause physiological changes in receivers[16]. For example, the releaser sex pheromone of burying beetles, *Nicrophorus vespilloides*, attracts mates[17], whereas the primer pheromone of queen bees, such as *Lasioglossum malachurum*, suppresses ovarian development in workers[18]. Traditionally, sex pheromones were deemed to be chemical signals between females and males, and females were thought to not sense their own sex pheromone[15,19]. Consequently, female behavior in response to female pheromones has rarely been studied[20]. However, it is now known that females of at least some insect species do sense, and respond to, their species-specific sex pheromone[20]. For example, females of the cotton bollworm, *Heliothis armigera*, the corn earworm, *Helicoverpa zea*, and the Mediterranean flour moth, *Ephestia kuehniella*, all avoid or disperse from locations with pheromone-permeated air[21,22]. However, the proximate resources for which these females compete have not been empirically studied.

Ecological theory predicts that a complex social context invokes competition for prey and mates and may also result in trade-offs between mate attraction and safety from predation, particularly when predators eavesdrop on communication signals of their prey[12,23]. In spiders, relatively little is known to what extent a complex social context invokes predator defense mechanisms. Generally, aggregated animals in a complex social context are more likely than solitary animals to draw the attention of predators[1,24].

Female cobweb spiders are ideal models for studying the effects of perceived same-sex competition and risk of predation[25,26]. Cobwebs, like other spider webs, have three main functions: prey capture[27], mate attraction[28], and safety from potential predators such as spider-hunting wasps that respond to chemical cues from spider prey[29,30]. Aggregations of cobwebs in the same micro-location (high-web-density settings)[31–34] may subject female spiders to severe competition for prey (social selection) and mates (sexual selection) as well as high predation risk ('struggle for survival'; natural selection). Whether female spiders can sense and respond to their species-specific sex pheromone and use this ability to reduce prey and mate competition has never been investigated.

Cobwebs, despite their seemingly unorganized appearance, have highly functional architecture, 'designed' to address all the spider's needs. These needs, however, are ever-changing. For example, hungry spiders invest more in prey-capture silk (Fig. 1a) than do sated spiders[25]. Similarly, spiders in high-web-density settings with perceived competition for prey should invest heavily in silk for prey capture. Furthermore, spiders in high-web-density settings, with vast chemical cues for spider-hunting wasps to exploit, may perceive an increased risk of predation and thus fortify their webs' safety area. Web adjustment by spiders in response to perceived competition for prey and mates, as well as the risk of predation, can be measured by quantifying changes in web characteristics, such as the number of silken strands females produce for prey capture and safety. Moreover, perceived mate competition can be assessed by quantifying the amount of courtship-inducing contact pheromone components deposited on silk and by determining the rate of their breakdown into airborne mate-attractant pheromone components (Fig. 1b)[26].

In our study, we used the false black widow spider, *Steatoda grossa*, as the model species. This solitary spider commonly dwells in buildings[35–37], where females build cobwebs with architectural characteristics resembling those of black widow webs[38]. To attract mates, *S. grossa* females deposit onto their webs three serine ester contact pheromone components (*N*-4-methylvaleroyl-*O*-butyroyl-L-serine (**1**), *N*-4-methylvaleroyl-*O*-isobutyroyl-L-serine (**2**) and *N*-4-methylvaleroyl-*O*-hexoyl-L-serine (**3**)) which then hydrolyze at their ester bonds, thereby releasing three corresponding mate-attractant pheromone components: butyric acid (**4**), isobutyric acid (**5**), and hexanoic acid (**6**)[26]. Concurrently, the serine amide breakdown product, *N*-4-methylvaleroyl-L-serine (**7**), accumulates on the webs (Fig. 1a)[26]. The transition—or breakdown—of contact pheromone components to mate-attractant pheromone components is thought to be mediated by a pH-dependent enzyme, with female spiders apparently able to manipulate the breakdown rate and thereby the attractiveness of their webs to mate-seeking males[26].

We predicted that *S. grossa* females can sense their social context, such as their presence in low- or high-web-density settings of conspecific females, and that they adjust their web in accordance with their perceived social context. As high-web-density settings likely come with strong same-sex competition for mates and prey, and with a high risk of predation, females would benefit from alleviating adverse effects related to competition and predation risk. Within this theoretical framework, we tested three hypotheses (H): (1) females in high-web-density settings adjust their webs to increase prey capture and lower predation risk; (2) females sense same-sex competition via airborne mate-attractant pheromone components; and (3) females in high-web-density settings increase their investment in mate attraction.

## Results

### H1: Females in high-web-density settings adjust their webs to increase prey capture and lower predation risk

*Experiment 1: Web adjustments by spiders in response to sequential exposure to low- and high-web-density settings.* Females in groups of three ($N = 16$) sequentially exposed to low- and high-web-density settings of conspecific females almost doubled their prey capture- and

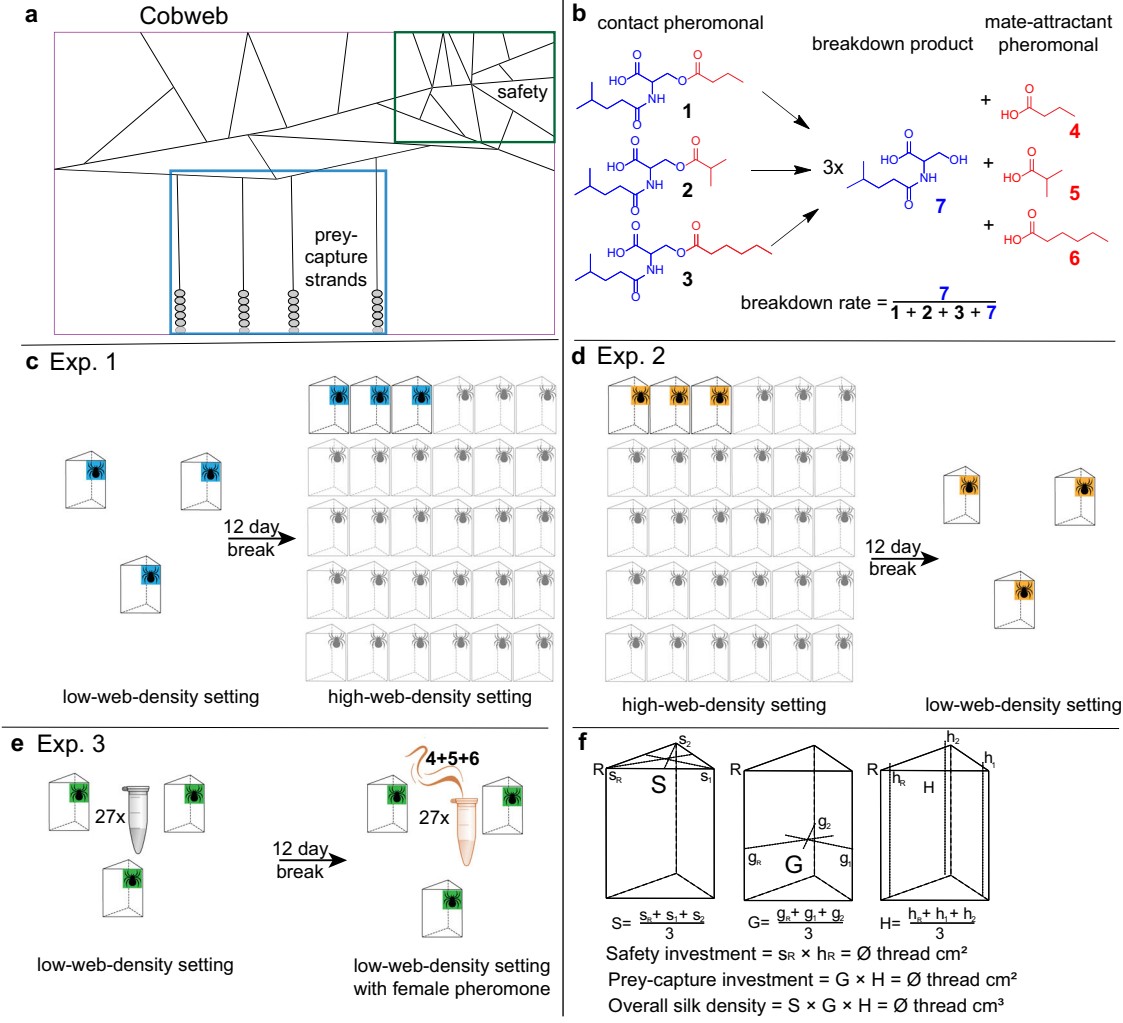

**Fig. 1 Graphical illustrations of a cobweb and experimental designs. a** Cobweb depicting the safety (retreat) section (green square) with numerous silk strands and glue-impregnated prey-capture lines (blue square) anchored to the ground. **b** Pheromone components of female *Steatoda grossa*: three serine ester contact pheromone components [*N*-4-methylvaleroyl-*O*-butyroyl-L-serine (**1**), *N*-4-methylvaleroyl-*O*-isobutyroyl-L-serine (**2**) and *N*-4-methylvaleroyl-*O*-hexoyl-L-serine (**3**)] prompt courtship by males, hydrolyze at the ester bond, and give rise to three corresponding mate-attracting acid pheromone components (red) [butyric (**4**), isobutyric (**5**), hexanoic (**6**)], while the serine amide breakdown product (blue), *N*-4-methylvaleroyl-L-serine (**7**), remains and accumulates on the web. The rate of the hydrolysis breakdown determines the web's attractiveness to males. **c** Design of experiment 1: Three female *S. grossa* built their webs for 48 h on three separate 3-dimensional frames (low-web-density setting); after a 12-day intermission, the same three females built their webs together with 27 other females (high-web-density setting). **d** Design of experiment 2: three females first built their webs in a high-web-density setting and, after a 12-day intermission, built webs in a low-web-density setting. **e** Design of experiment 3: three females built their webs first in a low-web-density setting and, after a 12-day intermission, built webs in the same low-web-density setting but permeated with synthetic mate-attracting pheromone components (4, 5, 6) at a concentration equivalent to a high-web-density setting. Pheromone components were formulated in mineral oil and released from 27 Eppendorf vials; during the first exposure, Eppendorf vials contained only plain mineral oil. **f** Web measurements were taken with a thin metal rod marked in 1-cm intervals[70] by recording the number of silken strands touching the rod in each interval. The rod was placed either vertically 1 cm away from the vertex of the triangular prism in the retreat corner ($h_R$) and the non-retreat corners ($h_1$, $h_2$) of the web or horizontally at the top of the retreat corner ($S_R$) and the non-retreat corners ($s_1$ and $s_2$) of the triangular prism, pointing to the center of the respective hypothenuses. Similar horizontal measurements were taken at the halfway-height point of the lateral edges.

safety-related silk investments (prey capture: $\chi^2 = 14.29$, df $= 1$, $p < 0.001$; safety: $\chi^2 = 15.93$, df $= 1$, $p < 0.001$; Exp. 1, Fig. 2a, b). The overall web density tripled with the transition to a more competitive social setting ($\chi^2 = 23.41$, df $= 1$, $p < 0.001$, Exp. 1, Fig. 2c). Across all groups, three spiders were lost between exposures and were excluded from data analyses.

*Experiment 2: Web adjustments by spiders in response to sequential exposure to high- and low-web-density settings.* Females in groups of three ($N = 16$) sequentially exposed to high- and low-web-density settings of conspecific females decreased their prey capture- and safety-related silk investments by ~40% (prey

capture: $\chi^2 = 20.53$, df $= 1$, $p < 0.001$; safety: $\chi^2 = 13.62$, df $= 1$, $p < 0.001$; Exp. 2, Fig. 2d, e). The overall silk investment decreased by almost half with the transition to a less competitive setting ($\chi^2 = 12.85$, df $= 1$, $p < 0.001$; Exp. 2, Fig. 2f). Across all groups, three spiders were lost between exposures and were excluded from data analyses.

**Hypothesis 2: Females sense same-sex competition via airborne mate-attractant pheromone components**
*Experiment 3: Web adjustments by spiders in response to sequential exposure to a low-web-density setting and to synthetic pheromone at a concentration mimicking a high-web-density*

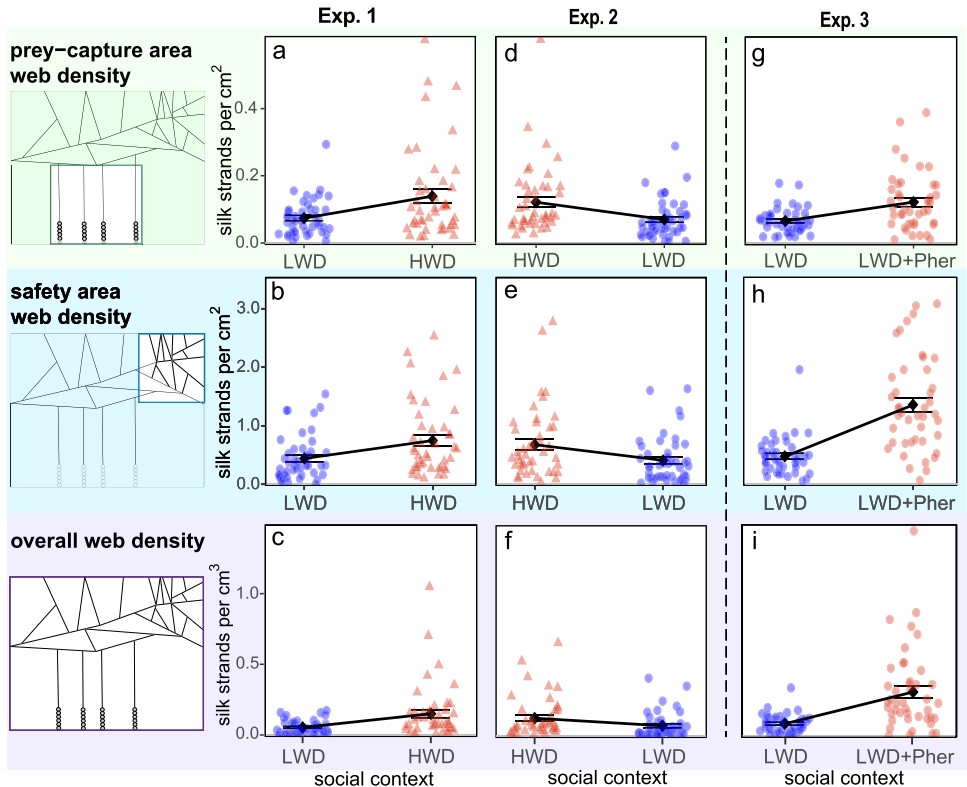

**Fig. 2 Web adjustments by female false black widow spiders in response to (1) changes in social context (Exps. 1, 2), and (2) synthetic sex pheromone indicating social-context change (Exp. 3).** In Exp. 1, when groups of three test spiders each ($N = 16$) first built their webs in a low-web-density setting ('LWD'; three test spiders only), and then rebuilt their webs in a high-web-density setting ('HWD'; three test spiders together with 27 further spiders) (see Fig. 1), the groups of test spiders rebuilding their webs produced more silk strands for prey capture (**a**), safety (**b**), and overall (**c**). In Exp. 2, when the 3-female groups ($N = 16$) first built their webs in an HWD setting and then rebuilt their webs in an LWD setting, they produced fewer silk strands for prey capture (**d**), safety (**e**), and overall (**f**). In Exp. 3, when the 3-female groups ($N = 16$) first built their webs in an LWD setting and then rebuilt their webs in an LWD setting while sensing synthetic pheromone at a concentration equivalent to an HWD setting ('LWD+Pher'), the groups rebuilding their webs produced more silk strands for prey capture (**g**), safety (**h**), and overall (**i**). Blue dots and red triangles indicate data of experimental replicates, and black squares with whiskers represent the mean and standard error. Web adjustments in each subpanel were statistically significant ($p < 0.001$; GLMM).

*setting*. When females in groups of three ($N = 16$) were sequentially exposed to a low-web-density setting and to synthetic sex-attractant pheromone components at a dose equivalent to that of a high-web-density setting, they adjusted their web as if they were exposed to a high-web-density setting, indicating recognition of social context based on the presence and concentration of airborne pheromone components. Silk investment in prey capture increased by 85% ($\chi^2 = 26.10$, df = 1, $p < 0.001$; Exp. 3, Fig. 2g), while silk investment in safety increased by 185% ($\chi^2 = 100.15$, df = 1, $p < 0.001$; Exp. 3, Fig. 2h). The overall web-density almost quadrupled ($\chi^2 = 82.67$, df = 1, $p < 0.001$, Exp. 3, Fig. 2i). Five spiders were lost between exposures and were excluded from data analysis.

**H3: Females in high-web-density settings increase their investment in mate attraction.** Females in groups of three ($N = 16$) sequentially exposed to low- and high-web-density settings increased (49%) the amount of contact pheromone components they deposited on their webs ($\chi^2 = 16.44$, df = 1, $p < 0.001$; Exp. 1, Fig. 3a). Conversely, females sequentially exposed to high- and low-web-density settings decreased (57%) the amount of contact pheromone components they deposited on their webs ($\chi^2 = 33.87$, df = 1, $p < 0.001$; Exp. 2, Fig. 3b). Females sequentially exposed to a low-web-density setting and to synthetic mate-attractant pheromone components at a dose equivalent to that of a high-web-density setting (1) increased (69%) the amount

of contact sex pheromone components they deposited on their webs ($\chi^2 = 23.79$, df = 1, $p < 0.001$, Exp. 3, Fig. 3c), and (2) increased (60%) the breakdown rate of contact pheromone to sex-attractant pheromone components ($\chi^2 = 5.28$, df = 1, $p = 0.022$; Exp. 4, Fig. 3d), essentially increasing their investment in mate attraction.

## Discussion

Our data show that female *S. grossa* sense the presence of other females via their pheromones and that they respond to perceived information by modifying both their own pheromone production and the architecture of their webs. Our findings imply that females relate the concentration of airborne conspecific pheromone to competition for mates and prey, and to the risk of predation.

Solitary web-building spiders can occur in large aggregations[31,34,39–42] that seem to present specific benefits to aggregation members, prompting them to remain in aggregations. For example, female western black widows, *Latrodectus hesperus*, were reluctant to leave aggregations even when their webs were severely disturbed, and unestablished spiders delayed relocation to new microhabitats when webs of conspecifics were present[31]. These data suggest that the benefits of staying together in a suitable microhabitat outweigh the costs of relocation, such as travel costs, mortality risk, and failure to find a new habitat[43,44]. The presence of established conspecifics in a microhabitat may

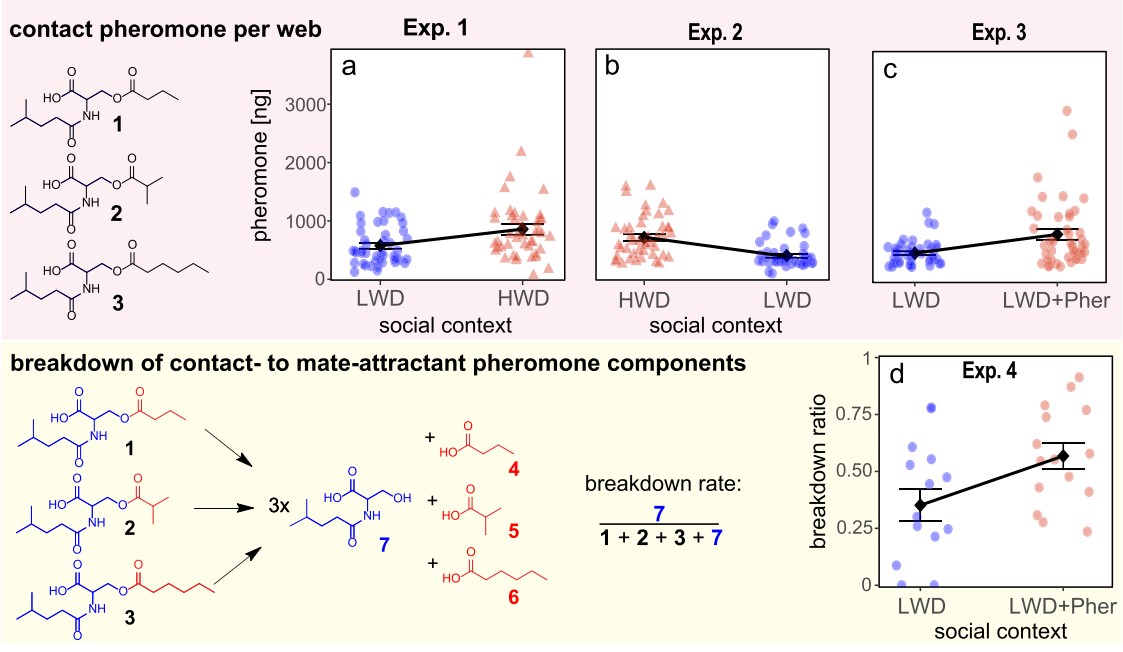

**Fig. 3 Adjustments for mate attraction by female false black widow spiders in response to perceived mate competition.** When groups of three test spiders each (N = 16) first built their webs in a low-web-density setting ('LWD'; three test spiders only), and then **a**, **b** rebuilt their webs in a high-web-density setting ('HWD'; three test spiders together with 27 further spiders), or **c** rebuilt their webs in a low-web-density setting while sensing synthetic pheromone at a concentration equivalent to a high-web-density setting ('LWD+Pher') (see Fig. 1), the groups of test spiders rebuilding their webs deposited more contact pheromone components [N-4-methylvaleroyl-O-butyroyl-L-serine (**1**), N-4-methylvaleroyl-O-isobutyroyl-L-serine (**2**) and N-4-methylvaleroyl-O-hexoyl-L-serine (**3**)] on their webs (upper row), and **d** accelerated their breakdown to mate-attractant pheromone components (red), [butyric acid (**4**), isobutyric acid (**5**), hexanoic acid (**6**)], essentially enhancing their webs' attractiveness to mate-seeking males. The serine amide breakdown product (blue), N-4-methylvaleroyl-L-serine (**7**), accumulates on the web. The rate of the hydrolysis breakdown determines the web's attractiveness to males. Blue dots and red triangles indicate data of experimental replicates, and black squares with whiskers represent the mean and standard error. Note changes in the amounts of contact pheromone components deposited on webs (Exps. 1–3; each $p < 0.001$, GLMM), and in the breakdown rate of contact pheromone components to mate-attractant pheromone components (Exp. 4; $p = 0.022$, GLMM), in response to perceived mate competition.

provide social information about habitat quality and prey availability, and may help save costs and time for habitat assessment[33,39,45]. Also, relocating and rebuilding a new web elsewhere is energetically costly to cobweb spiders which—unlike orb-weaving spiders—do not recycle their silk[43,46].

Group-living, however, has potential trade-offs. Spiders that have settled in groups will draw greater attention to predators—such as spider-hunting wasps—that exploit the chemical cues of their spider prey[28,29]. Conversely, there is safety in numbers in that the *per capita* predation risk may be 'diluted' in aggregations[1,47]. Similarly, female spider webs in aggregation will likely emanate more mate-attractant pheromone than single webs and thus be more attractive to males[34]. However, group-living females must then compete with each other for access to these prospective mates. Finally, although webs in aggregations may collectively capture more prey, prey capture for each individual female spider may suffer.

To study how female spiders respond to social context, such as their presence in a high-web-density setting with all its trade-offs for mate and prey competition as well as predation risk, we worked with the false black widow spider, *S. grossa*, a phylogenetic close relative of *L. hesperus*[48]. With the pheromone known[26], we could experimentally test whether *S. grossa* females sense their social setting via airborne mate-attractant pheromone components. The very nature of the 3-dimensional *S. grossa* web, with distinct safety and prey capture sections[25,38], further allowed us to assess whether spiders adjust their webs to alleviate adverse effects related to competition for prey and predation risk.

Our experimental data support the hypothesis that female *S. grossa* in high-web-density settings adjust their webs to increase prey capture and lower predation risk. We tested this hypothesis by sequentially exposing spiders to low- and high-web-density settings, and vice versa, allowing the same spiders to build their webs in each type of setting. Measuring the resulting web characteristics revealed that the spiders adjusted their webs in accordance with changes in social context. Progressing from low- to high-web-density settings with stronger competition for prey and predation risk, female spiders produced more silk strands for prey capture and fortified their webs' safety sections (Fig. 2). Conversely, progressing from high- to low-web-density settings with lower competition for prey and risk of predation, spiders curbed their silk production (Fig. 2), obviously saving energy. All data combined clearly indicate that the experimental spiders were aware of their social context. While it is well established that animals gauge habitat suitability, in part, by the presence or absence of conspecifics[49], it was not known that web-building spiders can sense the presence of female conspecifics via their sex pheromones and adjust their webs in relation to the perceived level of competition, and the threat of predation by natural enemies. Previously, sex pheromones of web-building female spiders were known in the context of sexual communication, attracting mates[50,51], and expediting the sexual maturation of subadult males[52,53].

Olfaction is the sensory modality underlying the detection of intra-sexual competitors. With webs being physically well separated in our experiments (Fig. 1), and with these web-building spiders deemed not to have good vision[54,55] and not to produce

sounds[55], we predicted airborne chemicals (sex pheromones) to be the signals or cues revealing the presence of intra-sexual competitors. Spiders—including widow spiders—do produce sex pheromones, of which some have been identified[26,28,51,56]. Although female-produced sex pheromones primarily serve in sexual communication to attract males[28,57], it was conceivable that female spiders can sense their species-specific pheromone (pheromone autodetection) and use this ability to gauge intra-sexual competition. To date, pheromone autodetection is known for a few insect species[20] and is believed to reduce mate competition. Pheromone autodetection by spiders was suggested[51] but not experimentally tested. With the female *S. grossa* sex pheromone available in our lab, we could test whether it serves a role in intra-sexual signaling among females. To study this question, we allowed three female spiders to build their webs in a confined room, then removed their webs, and—after a 12-day intermission—allowed the same three spiders to re-build their webs in the same room but permeated with synthetic mate-attractant pheromone at a concentration indicative of a high-web-density setting. Our findings that these pheromone-exposed spiders rebuilt their webs with enhanced prey capture and safety functions support the hypothesis of pheromone autodetection by female *S. grossa*. The pheromone receptor(s), however, remain unknown[58].

Female *S. grossa* responded to perceived mate competition by (1) depositing greater amounts of contact pheromone components on their webs (Fig. 3a–c) and (2) accelerating their breakdown to mate-attractant pheromone components (Fig. 3d), essentially increasing their webs' attractiveness to mate-seeking males. The mechanisms underlying this chemical breakdown are not fully understood, but there is convincing evidence that direct saponification alone is insufficient to explain the observed breakdown rates[26]. Instead, a web-borne carboxyl ester hydrolase enzyme, which is present on webs of *S. grossa*[26] and *L. hesperus*[59], is deemed responsible for the breakdown of contact pheromone components to mate-attractant pheromone components[26]. The concept is appealing because enzyme activity is pH-dependent[60], and spiders can manipulate the pH of their silk[61]. This enzyme concept could be experimentally tested by altering the webs' pH and by studying the pH-dependent enzymatic pheromone breakdown. Alternatively, synthetic contact pheromone components could be exposed to synthetic enzyme in different pH milieus, and the resulting pheromone breakdown rates could be measured. Regardless of the outcome of these experiments, our data (Fig. 3) indicate that female *S. grossa* do manipulate their webs' attractiveness in response to perceived mate competition. These results are intriguing because, to date, timed pheromone production and dissemination are known only in insects[62]. Although we demonstrate that female spiders adjust the pheromone titer on their webs, and thus their webs' attractiveness, according to perceived social context, we did not experimentally demonstrate but only infer the preferential attraction of males to webs constructed in high-web-density settings. This inference, however, is supported by a field study with *L. hesperus*, reporting that clusters of female webs were more attractive to mate-seeking males than isolated webs[34]. Males of *L. hesperus* arrived sooner, and at shorter time intervals, at clustered webs, likely because these webs release more pheromones and thus attract males over a wider range than isolated webs.

Our study demonstrates that aggregations of *S. grossa* webs represent a complex social context that invoked intra-sexual competition among female *S. grossa*. Females competed with each other for access to both mates and prey, and thus were concurrently subject to sexual and social selections[5,8,9,23]. Sex pheromones are a key determinant of mate attraction and reproductive success in many animal taxa[15], and thus are subject to selection as a secondary sexual trait. In response to same-sex competition, female *S. grossa* disseminated more mate-attractant pheromone from their webs (Fig. 3d) and thus became more apparent to mate-seeking males. Concurrent resource competition among female *S. grossa* was evident in web adjustments, such as the production of additional gum threads (Figs. 2a, d and 3a), which increased the likelihood of prey capture. Retaining only a few gum threads in the face of increased prey competition would incur missed-opportunity costs, forgoing the potential benefits of capturing more prey[1]. Conversely, in the less competitive setting of only three spiders, females produced fewer gum threads (Fig. 2). This adjustment is adaptive because silk and glue production is energetically costly, requiring 4.5 cal for 1 mg of silk[43]. Moreover, unlike orb-weavers which mitigate silk production costs by consuming their old silk, cobweb-weaving spiders, such as *S. grossa*, do not recycle their silk, making silk investments particularly costly[63]. As well-fed animals typically have greater reproductive capacity[64], female *S. grossa* in web aggregations are under social selection pressure for reproductive resources such as access to prey. That hungry cobweb spiders increase their investment in gum-footed lines for improved prey capture reveals the need-dependent plasticity of web architecture[25,65].

The complex social context presented by *S. grossa* web aggregations not only invoked intra-sexual competition for prey and mates, but it also invoked predator defense responses, as indicated by web adjustments to fortify the webs' safety section (Figs. 2b, e and 3b) which then makes it harder for parasitoids to access a prospective spider host[30]. Indeed, the 3-dimensional architecture of cobwebs is thought to have evolved from 2-dimensional sheets as an anti-predator adaption, allowing the construction of a silk-fortified retreat[66]. Generally, animals in aggregations are more likely than solitary animals to draw the attention of predators[67], but aggregated animals reduce individual predation risk through a dilution effect such that the *per capita* risk of predation decreases with increasing group size[67]. That *S. grossa* females did not rely on the dilution effect as a predator escape mechanism but instead lowered predation risk by strengthening their webs' safety section (Figs. 2b, e and 3b) indicates that *S. grossa* females in a complex social context are under significant natural selection pressure for survival. This selection pressure is likely exerted by spider-hunting wasps that respond to chemical cues from spider prey[29], with spiders in aggregations likely being semiochemically more apparent, and thus more attractive, to predatory wasps than single spiders. This selection pressure may be enhanced by predatory birds, amphibians and other spiders[27], which may also eavesdrop on chemical cues from aggregated spider webs. If these predators were to eavesdrop specifically on sex pheromone signals of spiders, then pheromonal signaling by spiders would incur both biosynthetic costs and ecological costs, essentially enhancing the risk of falling prey[68]. If so, this would exert selection pressure to optimize the cost/benefit ratio of signaling[15].

In conclusion, our study adds to the scarce body of literature on female-female competition in animals[10]. We show that female *S. grossa* spiders can sense their own pheromone and use this ability to gauge social context. In a complex social context, female *S. grossa* increases their competitiveness for mates and prey by disseminating more pheromone from their webs and by enhancing their webs' prey capture function. Concurrently, they reduce predation risk by fortifying their webs' safety section. All data combined indicate that intra-sexual competition of female *S. grossa* generates sexual, social and natural selection pressures that—in this perfect model system—could be separately studied and quantified.

## Methods

**Spider rearing**. *Steatoda grossa* spiders used in experiments were adult offspring of females collected on the Burnaby campus of Simon Fraser University[69]. The spiders were reared in the insectary of the Burnaby campus at 22 °C at a reversed 12L:12D photo cycle. All containers housing spiders were fitted with a moist cotton ball to increase relative humidity. Juvenile spiders were kept in petri dishes (100 × 20 mm) and provisioned once a week with *Drosophila melanogaster* vinegar flies. Sub-adults were separated by sex and kept individually. Adult virgin females were transferred from Petri dishes to 300-mL clear plastic cups (Western Family, Tigard, Oregon, USA) and provisioned once a week with black blow flies, *Phormia regina*. For experiments, naïve adult virgin females were randomly selected from a colony consisting of more than 700 virgin females and more than 80 family lines.

**Web-building for web-density measurements**. Each female was placed on a triangular frame (18 × 18 × 18 × 25 cm) of bamboo skewers (Bradshaw International Inc., CA, USA) and allowed 48 h to build her web. Individual frames were separated by at least 30 cm to allow assessment of social context only via airborne pheromone and were set in water-filled trays to prevent the spiders from escaping. Webs to be used in experiments 1 and 2 (*n* = 16 each) were built in a room (3.4 × 3 × 3.2 m) at 22 °C under a reversed 12L:12D photo cycle[69], whereas experiment 3 (*n* = 16) was run in four rooms (2.4 × 4.6 × 4 m; 2.4 × 1.7 × 3.2 m; 2.4 × 2 × 3.2 m; 2.4 × 3.3 × 2 m). Spiders were fed before and after, but not during, web-construction. Female age was not controlled because experiments tested for effects of low- and high-web-density settings on the same focal females irrespective of their age.

**Web measurements**. Web measurements were taken with a thin metal rod marked in 1-cm intervals[70] by recording the number of silken strands touching the rod in each interval. Nine measurements were taken for each web (Fig. 1f). For the first three measurements, the rod was placed vertically 1 cm away from the vertex of the triangular prism in the retreat corner ($h_R$) and the non-retreat corners ($h_1$, $h_2$) of the web. The next three measurements were taken by placing the rod horizontally at the top of the retreat corner ($s_R$) and the non-retreat corners ($s_1$ and $s_2$) of the triangular prism, pointing to the center of the respective hypothenuses. The final three measurements were taken by placing the rod at the halfway point of the lateral edges ($g_R$, $g_1$ and $g_2$) to the center of the respective hypothenuse at the same height. Twenty-two counts were taken for the vertical measurements and 15 counts for horizontal measurements from each corner of the frame. A value of 1 was added to one count for each of the nine measurements to avoid multiplication with zero in the calculations.

Investment in the safety section was calculated by multiplying the mean of vertical measurements from the retreat corner $s_R$ by the mean of top horizontal measurements from the retreat corner $h_R$ (Fig. 1f). Prey-capture investment was quantified by multiplying the mean of the mean halfway horizontal measurements from the retreat and the non-retreat corners G by the mean of the mean vertical counts from the retreat and the non-retreat corners H (Fig. 1f). The overall silk density was assessed as the product of the means of mean top horizontal measurements S, mean halfway down horizontal measurements G, and mean vertical measurements H (Fig. 1f).

**H1: Experiment 1: Web adjustments by spiders in response to sequential exposure to low- and high-web-density settings (Spring 2019)**. Groups (*N* = 16) of three virgin females were first exposed to a low-web-density setting (three web-building female spiders in the same room (3.4 × 3 × 3.2 m)) and allowed 48 h to build their webs on frames. Thereafter, these females were removed from

the frames, and web-density measurements were taken. Following a 12-day intermission, the same three females were placed in a high-web-density setting (30 web-building females in the same room) and allowed 48 h to build their webs (Fig. 1c). These three females were then removed from the frames, and web-density measurements were taken. Silk was collected from each frame with a glass rod (0.5 × 17.5 cm) and extracted in methanol for 24 h[71] for chemical analysis. The number of webs (30) in the high-web-density setting was inspired by a chance encounter of 15 females on webs under a table (3 × 1 m) in a storage room (A.F. person. obs.), whereas the number of webs (3) in the low-web-density setting was set 10-fold lower to achieve a significant numerical difference between the two web-density settings. The 12-day intermission between web constructions allowed females sufficient time to feed on provisioned prey and to replenish their nutrients for the energy-costly silk production.

**H1: Experiment 2: Web adjustments by spiders in response to sequential exposure to high- and low-web-density settings (Spring 2019)**. To control for potentially confounding sequential exposure effects, the order of exposure was reversed in experiment 2. Groups (*N* = 16) of three naïve females were first exposed to a high-web-density setting (30 web-building females in the same room (3.4 × 3 × 3.2 m)), and after a 12-day intermission, were exposed to the low-web-density setting, consisting of the three test spiders in a room (Fig. 1d). Webs were obtained, measured, and extracted as previously described.

**H2: Experiment 3: Web adjustments by spiders in response to sequential exposure to a low-web-density setting and to synthetic pheromone at a concentration mimicking a high-web-density setting (Summer 2019)**. To test whether females detect the numerical web density of conspecific females based on mate-attractant pheromone components, we modified the design of experiment 1. We used synthetic pheromone, instead of 30 female spiders, to purport a high-web-density setting. Groups (*N* = 16) of three naïve virgin females were first exposed to the low-web-density setting and were subsequently exposed to the same low-density setting but permeated with synthetic pheromone (**4, 5, 6**) at a concentration equivalent to a high-web-density setting[26]. Synthetic mate-attracting pheromone components were released from 27 400-µL Eppendorf vials, each containing **4** (0.112 µg), **5** (2.8 µg), and **6** (1.52 µg) dissolved in 200 µL of mineral oil. Any effects of the mineral oil were controlled by adding 27 Eppendorf vials containing plain mineral oil to the low-web-density setting (Fig. 1e). Each Eppendorf vial was perforated with a single hole using a No. 3 insect pin.

**H3: Quantification of contact pheromone components (Summer 2019)**. Increased investment in mate attraction was measured by quantifying the amount of contact pheromone components females deposit on their web and by calculating the rate of contact pheromone component breakdown into sex-attractant pheromone components.

Potential adjustments in the amount of contact sex pheromone components deposited by females on their webs in response to sequential exposure to (1) low- and high-web-density settings or vice versa (Exps. 1 & 2), or (2) to a low-web-density setting followed by exposure to synthetic pheromone at a concentration equivalent to a high-web-density setting (Exp. 3), were analyzed following established procedures[26]. Briefly, each web measured in experiments 1–3 was removed from its frame and then extracted 24 h in methanol (50 µL, 99.9% HPLC grade, Fisher Chemical, Ottawa, Canada). To avoid clogging the analytical instruments, web extracts were pre-purified using a Waters 600 high-performance liquid chromatograph (HPLC, Waters Corporation, Milford, MA, USA; 600

Controller, 2487 Dual Absorbance Detector, Delta 600 pump) fitted with a Synergy Hydro Reverse Phase C18 column (250 mm × 4.6 mm, 4 microns; Phenomenex, Torrance, CA, USA) which was eluted with isocratic acetonitrile (99.9% HPLC grade, Fisher Chemical, Ottawa, Canada) at 1 mL/min. The pheromone-containing fraction (3.00–4.4 min) was collected and concentrated to 1 mL. For pheromone quantification, aliquots (2 μL) were injected into a Bruker maXis Impact Quadrupole Time-of-Flight LC/MS System comprising an Agilent 1200 HPLC and a Bruker maXis Impact Ultra-High Resolution tandem TOF (UHR-Qq-TOF) mass spectrometer. The Agilent HPLC was fitted with a spursil C18 column (30 mm × 3.0 mm, 3 microns; Dikma Technologies, Foothill Ranch, CA, USA) which was heated to 30 °C and eluted with a solvent gradient (0.4 mL/min), starting with 80% water and 20% acetonitrile, and ending—after 4 min—with 100% acetonitrile. The solvent system contained 0.1% formic acid to enhance the peak shape of compounds. The mass spectrometer was set to positive electrospray ionization (+ESI) with a gas temperature of 200 °C and a gas flow of 9 L/min. The nebulizer was set to 4 bar and the capillary voltage to 4200 V. The major pheromone component **1** (ion 296, M+Na) was selected as a representative pheromone component. A calibration curve was established using synthetic **1** at 5 ng/μL, 2.5 ng/μL, 0.5 ng/μL, 0.25 ng/μL and 0.05 ng/μL.

**H3: Experiment 4: Calculation of breakdown rate of contact pheromone components to mate-attractant pheromone components (Spring 2022).** Groups ($N = 15$) of three naïve virgin females were first exposed to a low-web-density setting and then to the same low-web-density setting but permeated with synthetic pheromone at a concentration equivalent to a high-web-density setting. The webs of the three spiders per replicate were pooled, extracted in acetonitrile, and analyzed by HPLC-MS without prior purification. Contact pheromone components **1**, **2**, and **3**, as well as the breakdown product **7**, were quantified, and the breakdown ratio was calculated by dividing **7** by the sum of the **7** + **1** + **2** + **3** (Fig. 1b).

**Statistics and reproducibility.** Data were analyzed statistically using R[72]. Data on web architectural elements, as well as amounts of contact pheromone components deposited on webs, were analyzed for effects of social context using a repeated measures design. Webs from each group of three spiders were measured twice ($N = 16$ each, Exps. 1–3). We used generalized linear mixed models with tweedie family function of the glmmTMB package[73] to account for repeated measures of spiders within each group. A Type III ANOVA of the 'car' package (Anova) was used to test for the significance of social-context effects (low- or high-web-density setting) on response variables[74]. Model assumptions were checked using the DHARMa package[75]. Breakdown rates of contact pheromone components to mate-attractant pheromone components in relation to social context were quantified using a linear mixed effects model to account for the repeated measures of each group ($N = 15$, Exp. 4).

**Reporting summary.** Further information on research design is available in the Nature Portfolio Reporting Summary linked to this article.

## Data availability

Source data are available in Supplementary Data 1. All other data supporting the findings of this study are available within the paper and its Supplementary Information.

## Code availability

Code to analyze the data is available in Supplementary Data 2.

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

## Acknowledgements

We thank Dr. Kuntner and two anonymous reviewers for constructive comments, Jaime-Lynne Varney and Emmanuel Hung for assistance during preliminary studies, and Hongwen Chen, Sharon Oliver, Hanna Watkins and Em Lim for technical advice or comments. A.F. was supported by Graduate Fellowships from SFU, the H.R. McCarthy Bursary, and an Alexander Graham Bell Scholarship from the Natural Sciences and Engineering Research Council of Canada (NSERC). The project was funded by an NSERC–Industrial Research Chair to G.G., with Scotts Canada Ltd. and BASF Canada Inc. as the industrial sponsors. The funders had no role in study design, data collection and analysis, decision to publish, or preparation of the manuscript.

## Author contributions

Conceptualization: A.F. and G.G.; Methodology: A.F. and G.G.; Investigation: A.F., Y.F., A.P., and S.M.-D.S.; Data curation: A.F.; Writing—original draft: A.F. and G.G.; Writing—review & editing: G.G. and A.F.; Funding acquisition: G.G.; Resources: G.G., A.F.; Supervision: A.F.

## Competing interests

The authors declare no competing interests.
