## [Peer Review File · Communications Biology]

Reviewers' comments:

Reviewer #1 (Remarks to the Author):

This paper reports on the original experimental work on the spider *S. grossa* and the authors have shown through a set of carefully designed experiments that the female spiders sense intra-sexual competition for prey and mates, and predation risk, via airborne mate-attractant pheromone components. In response to perceived intra-sexual competition, female spiders adjusted their webs to increase prey capture and lower predation risk and increased their investment in mate attraction. This is an elegant study that in my view needs to be published with minimum delay. I find no factual errors, but need to voice one concern that relates to the narrative. While female adjustments are clearly in response to levels of competition from other females, this is a textbook case of intraspecific competition, but labeling this as "intra-sexual conflict" may not be warranted. I'd ask the authors to reconsider these semantic issues not to introduce unneeded dramatization to the otherwise brilliant narrative.

M. Kuntner

Reviewer #2 (Remarks to the Author):

The authors studied female-female conflict of cob-web spider *Steatoda grossa* using the sex pheromones as a proxy. The study is novel, and will provide a valuable reference for studies of female-female conflicts in animals. However, I have several major concerns which I think should be addressed before it can be considered to publish in *Communications Biology*.

1. The authors tried to set up a connection between web construction behaviors and the doses of sex pheromone, however, they did not test the real attraction effects of those experimental settings to males. To attract males, a threshold dose may take effect, so it is highly possible that there is no significant difference between the male lure effect among all the experimental settings the authors used. Similarly, without behavioral evidence of predator avoidance of different settings, the connection between web construction behavior and dose usage would be weak.

2. The pictures are messy as well as the figure legends, very tedious. For example, Figure 1a, Figure 2a and Figure 3a are repetitious without novelty. Also, the authors may use a real picture of the cobweb to show what it looks like in the field, not just the graphical illustrations. It takes too much space to describe the experimental design differences between Figure 1c and 1d. In addition, it is better to present the statistical result in the text rather than in the figure legend, which should be very clean, no citation, no discussion, no method-description. Besides, the legend of Figure 2 showed GLMM analysis result, but it was absent in 'Statistical analyses' part. The chi-square test was not mentioned either.

3. One of the hypotheses the authors tested is that females in high-web-density settings adjust their webs to increase prey capture and lower predation risk, however, there are a lot of similar studies in spiders, such as Zhang et al., (2022), Blackledge and Zevenbergen, 2007. The authors also reviewed it in Line 83-88. Thus, the novelty of this part is not outstanding.

Blackledge TA, Zevenbergen JM, 2007. Condition-dependent spider web architecture in the western black widow, *Latrodectus hesperus*. *Animal Behaviour* 73:855-864. doi: 10.1016/j.anbehav.2006.10.014.

Zhang H, Li G, Li C, Chen J, Zhao Z, Zhang S, Liu J, 2022. Feeding mediated web-building plasticity in a cobweb spider. *Current Zoology:zoac077*. doi: 10.1093/cz/zoac077.

Reviewer #3 (Remarks to the Author):

I really enjoyed reading this well conducted study on how female web spiders respond to same sex competition via their sensing of pheromones from female conspecifics. Careful experimentation with impressive replication has revealed highly relevant data on female spider behaviour in the context of same-sex competition. The paper is overall very well written, although I found a few sentences that may benefit from a rewording. Mainly in the introduction I stumbled over wordings that are not quite correct or could be misunderstood. Below is a list of these minor queries with line numbers. I suggest that the authors carefully check the introduction and discussion again for wordings and grammar.

Line 37: sentence can be misunderstood: it is not the "selection of traits" that is shaped but the trait

Line 38-39: questionable claim? - as females are in the end the selective force on males which evolve all these traits to impress them; you initially make this sound as if people did not look at female coloration, adornment and the like in females because they were biased. Perhaps females are simply - as you say in line 41 - less often adorned etc.

Line 49: is it justified to call such a weaponry a sexual trait?

Line 74: are you not rather looking at the trade-off between mate-attraction and predator attraction?

Line 71: I am not sure I buy this claim; there seems to be a lot of research on anti-predatory strategies, non-consumptive effects and the like.

Line 81: rather call it "their species-specific pheromone" than their "own"?

Line 185: I do not understand that sentence. What has safety in numbers to do with the web architecture?

Line 194-196: Why do you need to justify that you did not use *L. hesperus*?

Line 214: this claim may not be right as several studies have for example shown that subadult male spiders speed up their development in the presence of female pheromones

For my liking, the neglect of research on female competition is slightly overstated. I do agree that there are not many examples on female competition over males and one reason is that such species are rare and another one might be that less obvious modalities are at play – such as the production of chemical signals. Web building spiders combine both of these conditions since females are sedentary and frequently aggregated and chemical communication is the main mode in the context of mate attraction. Spiders are indeed understudied and very little is known about the strategic use of pheromones in general and particularly in spiders. A recent paper has shown for the first time that females that remain unmated for a certain period will increase their pheromone production and thereby increase their attractivity for males. The present study adds another highly interesting component that influences pheromone quantity and that is the presence of other females. This is a very interesting, novel and important finding. My suggestion is to tone down the presumed bias against female to emphasize other arguments a bit more, as for example potential costs of chemical signalling and trade-offs between mate attraction and safety in species in which females are the signalling sex. However, this is surely a matter of taste.

In addition to measuring the pheromone production of 3 target females in every trial, the authors also assessed two web areas known to be relevant for foraging and for safety, respectively. Females added more silk to both these areas when they sensed competition. While the paper does not address the adaptive value of these density dependent investments, there are previous studies cited which provide suggestive evidence in this respect (which could be discussed a bit more .. see below). It is likely that more gum threads will increase prey capture success and that a dense retreat will make it harder for a parasitic wasp to get to the spider. It is less clear to me why females would produce fewer capture threads when among 3 conspecifics than among 30. Are they indeed so costly to produce? I suspect that it is known that females respond to scarcity of prey by adding more adding gum threads? A discussion of these issues would be nice.

I am not a fan of formats where the results come before the methods. It does require that some methods have to be stated twice to understand what has been done. While hypotheses and predictions are very clear, I was rather confused about the sample sizes and replication numbers and would

suggest to make very clear that you repeated each treatment 16 times already in the result section. The first sentence of the discussion is again phrased slightly incorrectly. It sounds odd that there is „intra-sexual competition for predation risk“. I suggest to rephrase this by saying that „females sense the presence of other females via their pheromones and respond to social information by increasing pheromone production, as well as modifying their webs These findings suggest that females relate the presence of conspecific pheromones to competition for mates and prey and to an increased risk of predation.

The discussion is comprehensive although I missed a discussion of the (potential) adaptive values behind the changes females made to the webs and pheromones.

Methods are well explained but I have a few questions:

How many different families did you raise in the lab and did you consider family background when assembling the groups?

Did you feed the females before, during and between the trials?

Why did you wait 12 days between the repeated trials?

Maybe you could also provide a reason why you used 30 females in the high-density treatment. Is this resembling a realistic density in nature? I was also wondering about the distance between webs, which I also missed a justification for.

Argiope bruennichi females increase pheromone production with increasing adult age if they remain unmated. Do you have any information on age related pheromone production in the false widow? Did you take female age composition into account when assembling the groups and target females? Did you measure any age-related differences in pheromone production of your focal females? Will females eventually produce eggsacs if they are unmated?

Responses (R) to reviewers' Comments

Please note: all references to line numbers refer to the manuscript with tracked changes

Reviewers' comments:

Reviewer #1 (Remarks to the Author):

1. This paper reports on the original experimental work on the spider *S. grossa* and the authors have shown through a set of carefully designed experiments that the female spiders sense intra-sexual competition for prey and mates, and predation risk, via airborne mate-attractant pheromone components. In response to perceived intra-sexual competition, female spiders adjusted their webs to increase prey capture and lower predation risk and increased their investment in mate attraction. This is an elegant study that in my view needs to be published with minimum delay.

R1: We thank you, Dr. Kuntner, for your positive assessment of our manuscript and your kind remarks!

2. I find no factual errors, but need to voice one concern that relates to the narrative. While female adjustments are clearly in response to levels of competition from other females, this is a textbook case of intraspecific competition, but labeling this as "intra-sexual conflict" may not be warranted. I'd ask the authors to reconsider these semantic issues not to introduce unneeded dramatization to the otherwise brilliant narrative.

M. Kuntner

R2: Thank you for having pointed out this semantic distinction. We have revised the manuscript accordingly.

Reviewer #2 (Remarks to the Author):

3. The authors studied female-female conflict of cob-web spider *Steatoda grossa* using the sex pheromones as a proxy. The study is novel, and will provide a valuable reference for studies of female-female conflicts in animals. However, I have several major concerns which I think should be addressed before it can be considered to publish in *Communications Biology*.

R3: We thank Reviewer #2 for the overall positive assessment! We have strived to address all concerns.

4. The authors tried to set up a connection between web construction behaviors and the doses of sex pheromone, however, they did not test the real attraction effects of those experimental settings to males. To attract males, a threshold dose may take effect, so it is highly possible that there is no significant difference between the male lure effect among all the experimental settings the authors used.

R4: We agree with Reviewer #2 that we did not bioassay the behavioural responses of males. However, we did demonstrate that female spiders adjusted the pheromone titer on their webs, and thus their webs' attractiveness, according to perceived social context. A web's pheromone titer, in turn, affects the range over which the web is attractive to mate-seeking males. Demonstrating differential attraction of males to webs that females constructed in low or high web-density settings would have required to determine the attractive range of low and high web-density webs, and then to test for attraction of males to these webs at various distances. This would have been difficult to do in a laboratory setting, and would have been beyond the scope of this manuscript. To address this comment, we have expanded the discussion, citing a

study that demonstrated pheromone dose-dependent attraction of male spiders (lines 265-272): “Although we demonstrate that female spiders adjust the pheromone titer on their webs, and thus their webs’ attractiveness, according to perceived social context, we did not experimentally demonstrate, but only infer, preferential attraction of males to webs constructed in high web-density settings. This inference, however, is supported by a field study with *L. hesperus*, reporting that clusters of female webs were more attractive to mate-seeking males than isolated webs.³⁴ Males of *L. hesperus* arrived faster, and at shorter time intervals, at clustered webs, likely because these webs release more pheromone and thus attract males over a wider range than isolated webs.

5. Similarly, without behavioral evidence of predator avoidance of different settings, the connection between web construction behavior and dose usage would be weak.

R5 We agree. We did not demonstrate - in behavioral studies - that a fortified web safety section reduced the incidence of predation. However, to generate this type of data set would have been a monumental task way beyond the scope of this manuscript. To address this comment, we have revised a section in the discussion, as follows (lines 295-298): “The complex social context presented by *S. grossa* web aggregations not only invoked intra-sexual competition for prey and mates, but it also invoked predator defense responses, as indicated by web adjustments to fortify the webs’ safety section (Figs. 2b+e, 3b) which makes it harder for parasitoids to access a prospective spider host. Indeed, the 3-dimensional architecture of cobwebs is thought to have evolved from 2-dimensional sheets as an anti-predator adaption, allowing the construction of a silk-fortified retreat.⁷⁰

6. 2. The pictures are messy as well as the figure legends, very tedious. For example, Figure 1a, Figure 2a and Figure 3a are repetitious without novelty.

R6. To address this comment, we have merged Figures 2 and 3 into Figure 2. We contend that figure and figure caption are supposed to be a stand-alone unit, and in combination must provide sufficient information for readers. We assume there is a reason why ‘Communications Biology’ allows as many as 350 words in a figure caption.

7. Also, the authors may use a real picture of the cobweb to show what it looks like in the field, not just the graphical illustrations.

R7. As suggested, we have taken photographs of a cobweb but they do not sufficiently well depict all pertinent information we want to convey. In support of our conclusion, we would like to refer the reviewer to the cobweb photograph in Fig 1b in Scott et al. 2015 J Chem Ecol.

8. It takes too much space to describe the experimental design differences between Figure 1c and 1d.

R8. We contend that the graphical illustrations in Figure 1c and 1d are critically important to understand the experimental design of the entire study. Thus, we prefer not to compromise clarity by miniaturizing or simplifying these illustrations.

9. In addition, it is better to present the statistical result in the text rather than in the figure legend, which should be very clean, no citation, no discussion, no method-description.

R9. Again, we are ‘old school’ and consider a figure and figure caption a stand-alone unit. Please also see R6.

10. Besides, the legend of Figure 2 showed GLMM analysis result, but it was absent in ‘Statistical analyses’ part. The chi-square test was not mentioned either.

R10: We contend that we had listed in the appropriate sections of the review manuscript all statistical tests, including the GLMM and Type III ANOVA (which uses Chi rather than F statistics).

11. 3. One of the hypotheses the authors tested is that females in high-web-density settings adjust their webs to increase prey capture and lower predation risk, however, there are a lot of similar studies in spiders, such as Zhang et al., (2022), Blackledge and Zevenbergen, 2007. The authors also reviewed it in Line 83-88. Thus, the novelty of this part is not outstanding. Blackledge TA, Zevenbergen JM, 2007. Condition-dependent spider web architecture in the western black widow, *Latrodectus hesperus*. *Animal Behaviour* 73:855-864. doi: 10.1016/j.anbehav.2006.10.014. Zhang H, Li G, Li C, Chen J, Zhao Z, Zhang S, Liu J, 2022. Feeding mediated web-building plasticity in a cobweb spider. *Current Zoology:zoac077*. doi: 10.1093/cz/zoac077.

R11: We contend that Blackledge & Zevenbergen, as well as Zhang et al., demonstrate web adjustment in response to physiological condition and to diet availability. We, in contrast, studied web adjustment in response to social context which required perception of airborne pheromone by female spiders. This differs from the above-cited studies, and is truly novel. Nonetheless, we thank the Reviewer to pointing us to Zhang et al. which we cite in the revised manuscript (lines 291-292) “That hungry cobweb spiders increase their investment in gum-footed lines for improved prey-capture, reveals need-dependent plasticity of web architecture .^{25,65}”

Reviewer #3 (Remarks to the Author):

12. I really enjoyed reading this well conducted study on how female web spiders respond to same sex competition via their sensing of pheromones from female conspecifics. Careful experimentation with impressive replication has revealed highly relevant data on female spider behaviour in the context of same-sex competition.

R12: We thank Reviewer 3 for the positive feedback.

13. The paper is overall very well written, although I found a few sentences that may benefit from a rewording. Mainly in the introduction I stumbled over wordings that are not quite correct or could be misunderstood. Below is a list of these minor queries with line numbers. I suggest that the authors carefully check the introduction and discussion again for wordings and grammar.

R13: Thank you, again, for the complimentary comment. We trust that we have adequately addressed all comments and queries.

14. Line 37: sentence can be misunderstood: it is not the "selection of traits" that is shaped but the trait

R14: We agree. Revised accordingly (line 39).

15. Line 38-39: questionable claim? - as females are in the end the selective force on males which evolve all these traits to impress them; you initially make this sound as if people did not look at female coloration, adornment and the like in females because they were biased. Perhaps females are simply - as you say in line 41 - less often adorned etc.

R15: The claim was based on a review by Ah-King 2022 in ‘Nature Communications’. Nonetheless, we understand the reviewer’s concern, and have addressed the comment by revising the sentence to read (lines 41-43): “Studies of evolutionary selection for secondary sexual traits, such as marked coloration, large size, or striking adornments, have prioritized males over females, possibly because these traits are generally more apparent in males than in females.”¹⁰

16. Line 49: is it justified to call such a weaponry a sexual trait?

R16: As suggested, we have removed ‘secondary sexual’.

17. Line 74: are you not rather looking at the trade-off between mate-attraction and predator attraction?

R17: We have expanded the wording (lines 73-75): “Ecological theory predicts that a complex social context invokes competition for prey and mates, and may also result in trade-offs between mate attraction and safety from predation, particularly when predators eavesdrop on communication signals of their prey”.

18. Line 71: I am not sure I buy this claim; there seems to be a lot of research on anti-predatory strategies, non-consumptive effects and the like.

R18: We have rephrased the pertinent sentence (lines 75-76): “In spiders, relatively little is known to what extent a complex social context invokes predator defense mechanisms. Generally, aggregated animals in a complex social context are more likely than solitary animals to draw the attention of predators”.²⁴

19. Line 81: rather call it “their species-specific pheromone” than their “own”?

R19: Revised accordingly throughout the manuscript.

20. Line 185: I do not understand that sentence. What has safety in numbers to do with the web architecture?

R20: Thank you catching this editorial ‘left-over’! The revised sentence reads (line 195): “Conversely, there is safety in numbers in that the *per capita* predation risk may be ‘diluted’ in aggregations.”

21. Line 194-196: Why do you need to justify that you did not use *L. hesperus*?

R21: Good point! We have removed the respective sentence.

22. Line 214: this claim may not be right as several studies have for example shown that subadult male spiders speed up their development in the presence of female pheromones

R22: We have revised this section to read (lines 223-229): “While it is well established that animals gauge habitat suitability, in part, by the presence or absence of conspecifics,⁴⁹ it was not known that web-building spiders can sense the presence of female conspecifics via their sex pheromones, and adjust their webs in relation to the perceived level of competition, and threat of predation by natural enemies. Previously, sex pheromones of web-building female spiders were known in the context of sexual communication, attracting mates,^{e.g.50,51} and expediting sexual maturation of subadult males.^{52,53}

23. For my liking, the neglect of research on female competition is slightly overstated. I do agree that there are not many examples on female competition over males and one reason is that such

species are rare and another one might be that less obvious modalities are at play – such as the production of chemical signals. Web building spiders combine both of these conditions since females are sedentary and frequently aggregated and chemical communication is the main mode in the context of mate attraction. Spiders are indeed understudied and very little is known about the strategic use of pheromones in general and particularly in spiders. A recent paper has shown for the first time that females that remain unmated for a certain period will increase their pheromone production and thereby increase their attractiveness for males. The present study adds another highly interesting component that influences pheromone quantity and that is the presence of other females. This is a very interesting, novel and important finding. My suggestion is to tone down the presumed bias against female to emphasize other arguments a bit more, as for example potential costs of chemical signalling and trade-offs between mate attraction and safety in species in which females are the signalling sex. However, this is surely a matter of taste.

R23: We have de-emphasized the male bias in past studies (please see R15), mentioned the trade-off between mate attraction and safety from predation (please see R17), and further emphasize ecological aspects (lines 41-43; 73-75; 196; 227-229; 282-289; 291-292; 295-298; 309-312).

24. In addition to measuring the pheromone production of 3 target females in every trial, the authors also assessed two web areas known to be relevant for foraging and for safety, respectively. Females added more silk to both these areas when they sensed competition. While the paper does not address the adaptive value of these density dependent investments, there are previous studies cited which provide suggestive evidence in this respect (which could be discussed a bit more .. see below). It is likely that more gum threads will increase prey capture success

R24. We have expanded the pertinent sentence (line 280-284): “Concurrent resource competition among female *S. grossa* was evident in web adjustments, such as the production of additional gum threads (Figs. 2a+d, 3a), which increase the likelihood of prey capture. Retaining only few silk lines in the face of increased prey competition would incur missed-opportunity costs, forgoing potential benefit to capture more prey”.¹

25. and that a dense retreat will make it harder for a parasitic wasp to get to the spider.

R25. Again, we have expanded the pertinent sentence (line 293-298): “The complex social context presented by *S. grossa* web aggregations not only invoked intra-sexual competition for prey and mates, but it also invoked predator defense responses, as indicated by web adjustments to fortify the webs’ safety section (Figs. 2b+e, 3b) which then makes it harder for parasitoids to access a prospective spider host. Indeed, the 3-dimensional architecture of cobwebs is thought to have evolved from 2-dimensional sheets as an anti-predator adaption, allowing the construction of a silk-fortified retreat”.⁷⁰

26. It is less clear to me why females would produce fewer capture threads when among 3 conspecifics than among 30. Are they indeed so costly to produce? I suspect that it is known that females respond to scarcity of prey by adding more adding gum threads? A discussion of these issues would be nice.

R26. To provide an explanation, we have expanded the discussion (lines 280-289): “Concurrent resource competition among female *S. grossa* was evident in web adjustments, such as the production of additional gum threads (Figs. 2a+d, 3a), which increase the likelihood of prey capture. Retaining only few gum threads in the face of increased prey competition would incur missed-opportunity costs, foregoing potential benefits of capturing more prey. Conversely, in the less competitive setting of only three spiders, females produced fewer gum threads (Fig. 2). This adjustment is adaptive because silk and glue

production is energetically costly, requiring 4.5 cal for 1 mg of silk (Tanaka 1889). Moreover, unlike orb-weavers which mitigate silk production costs by consuming their old silk (Opell, 1998), cobweb-weaving spiders, such as *S. grossa*, do not recycle their silk making silk investments particularly costly”.

27. I am not a fan of formats where the results come before the methods. It does require that some methods have to be stated twice to understand what has been done. While hypotheses and predictions are very clear, I was rather confused about the sample sizes and replication numbers and would suggest to make very clear that you repeated each treatment 16 times already in the result section.

R27: Good point! Wherever indicated, we have added the number of replicates already to the Result section.

28. The first sentence of the discussion is again phrased slightly incorrectly. It sounds odd that there is „intra-sexual competition for predation risk“. I suggest to rephrase this by saying that „females sense the presence of other females via their pheromones and respond to social information by increasing pheromone production, as well as modifying their webs These findings suggest that females relate the presence of conspecific pheromones to competition for mates and prey and to an increased risk of predation.

R28: Thank you for the comment! We have rephrased the first sentence of the discussion to read (lines 174-178): “Our data show that female *S. grossa* sense the presence of other females via their pheromones, and that they respond to perceived information by modifying both their own pheromone production and the architecture of their webs. Our findings imply that females relate the concentration of airborne conspecific pheromone to competition for mates and prey, and to risk of predation”.

29. The discussion is comprehensive although I missed a discussion of the (potential) adaptive values behind the changes females made to the webs and pheromones.

R29: We have expanded the discussion in various places to reflect the adaptive value of altered pheromone production and modified web architecture in response to perceived social context. Please see specific responses R4, R15, R17, R22, R25, and R26.

30. Methods are well explained but I have a few questions:
How many different families did you raise in the lab and did you consider family background when assembling the groups?

R30: We raised a very large colony of *S. grossa* with more than 83 family lines. Over the course of this study, we had more than 700 virgin females available. We have added this information (lines 331-333): ‘For experiments, naïve adult virgin females were randomly selected from a colony consisting of more 700 virgin females and more than 80 family lines.

31. Did you feed the females before, during and between the trials?

R31: Good question! We have added the information (lines 343-344): “Spiders were fed before and after, but not during, web-construction.”

32. Why did you wait 12 days between the repeated trials?

R32: We have provided an explanation (lines 384-386): “The 12-day intermission between web constructions allowed females sufficient time to feed on provisioned prey and to replenish their nutrients for the energy-costly silk production”.

33. Maybe you could also provide a reason why you used 30 females in the high-density treatment. Is this resembling a realistic density in nature?

R33: We have provided an explanation (lines 380-384): “The number of webs (30) in the high-web-density setting was inspired by a chance encounter of 15 females on webs under a table (3×1 m) in a storage room (A.F., person. obs.), whereas the number of webs (3) in the low-web-density setting was set 10-fold lower to achieve a significant numerical difference between the two web-density settings”.

34. I was also wondering about the distance between webs, which I also missed a justification for.

R34: We have added the information and provided an explanation (lines 338-340): “Individual frames were separated by at least 30 cm to allow assessment of social context only via airborne pheromone, and were set in water-filled trays to prevent the spiders from escaping”.

35. *Argiope bruennichi* females increase pheromone production with increasing adult age if they remain unmated. Do you have any information on age related pheromone production in the false widow?

R35: We do have age-related information on pheromone production but these data are part of another study and will be presented in another manuscript currently in preparation. In this manuscript, we investigated the effect of age of virgin spider females on fecundity (life-time-reproductive output), deposition of contact pheromone on webs, and release of mate-attracting pheromone components. We expect to submit this manuscript by the end of the year.

36. Did you take female age composition into account when assembling the groups and target females? Did you measure any age-related differences in pheromone production of your focal females?

R36: We did not track or control for age of females because we compared the effect of low- and high-web-density settings on the same focal females. We have added an explanation (lines 344-345): “Female age was not controlled because experiments tested for effects of low- and high-web-density settings on the same focal females irrespective of their age”.

37. Will females eventually produce eggsacs if they are unmated?

R37: Virgin female *Steatoda grossa* do not produce egg sacs. Interestingly, however, their fecundity significantly decreases throughout their long adult life (Fischer et al. unpublished data). This will be presented and discussed in our next manuscript.

REVIEWERS' COMMENTS:

Reviewer #2 (Remarks to the Author):

The manuscript has been greatly improved, and has addressed all my concerns.

Reviewer #3 (Remarks to the Author):

Thank you for a careful revision